# Should Endometrial Cancer Treatment Be Centralized?

**DOI:** 10.3390/biology11050768

**Published:** 2022-05-18

**Authors:** Vincenzo Dario Mandato, Andrea Palicelli, Federica Torricelli, Valentina Mastrofilippo, Chiara Leone, Vittoria Dicarlo, Alessandro Tafuni, Giacomo Santandrea, Gianluca Annunziata, Matteo Generali, Debora Pirillo, Gino Ciarlini, Lorenzo Aguzzoli

**Affiliations:** 1Unit of Obstetrics and Gynecology, Azienda USL-IRCCS di Reggio Emilia, 42122 Reggio Emilia, Italy; valentina.mastrofilippo@ausl.re.it (V.M.); chiara.leone1991@gmail.com (C.L.); vittoria.dicarlo@ausl.re.it (V.D.); gianluca.annunziata@ausl.re.it (G.A.); matteo.generali@ausl.re.it (M.G.); debora.pirillo@ausl.re.it (D.P.); 2Pathology Unit, Azienda USL-IRCCS di Reggio Emilia, 42122 Reggio Emilia, Italy; andrea.palicelli@ausl.re.it (A.P.); alessandro.tafuni@ausl.re.it (A.T.); giacomo.santandrea@ausl.re.it (G.S.); 3Laboratory of Translational Research, Azienda USL-IRCCS di Reggio Emilia, 42122 Reggio Emilia, Italy; federica.torricelli@ausl.re.it; 4Pathology Unit, Department of Medicine and Surgery, University of Parma, 43121 Parma, Italy; 5Clinical and Experimental Medicine PhD Program, University of Modena and Reggio Emilia, 41121 Modena, Italy; 6Unit of Surgical Gynecol Oncology, Azienda USL-IRCCS di Reggio Emilia, 42122 Reggio Emilia, Italy; gino.ciarlini@ausl.re.it (G.C.); lorenzo.aguzzoli@ausl.re.it (L.A.)

**Keywords:** endometrial cancer, gynecologic oncologists, general gynecologists, centralization, lymphadenectomy, sentinel lymph node biopsy, frozen section, laparoscopy, high volume centers, tumor board

## Abstract

**Simple Summary:**

Endometrial cancer (EC) is the most frequent cancer of the female genital tract in Western and emerging countries. It is commonly thought to be easy to treat as it is usually diagnosed at an early stage, and hysterectomy with bilateral adnexectomy alone is the only treatment required. The progressive replacement of systematic lymphadenectomy with sentinel lymph node biopsy has also fostered this erroneous belief. In fact, if mistreated, EC has a high lethality due to its poor response to chemotherapy in the cases of advanced stage and recurrence. In this paper we reviewed the literature to understand whether the treatment of EC should be reserved for gynecologic oncologists and whether treatment of EC patients should be centralized in high-volume hospitals. We also evaluated the contribution of other specialists involved in the diagnostic and therapeutic pathways of EC. Finally, we suggest a possible network to treat EC patients to ensure effective staging and treatment.

**Abstract:**

Endometrial cancer (EC) is the most common malignancy of the female genital tract in Western and emerging countries. In 2012, new cancer cases numbered 319,605, and 76,160 cancer deaths were diagnosed worldwide. ECs are usually diagnosed after menopause; 70% of ECs are diagnosed at an early stage with a favorable prognosis and a 5-year overall survival rate of 77%. On the contrary, women with advanced or recurrent disease have extremely poor outcomes because they show a low response rate to conventional chemotherapy. EC is generally considered easy to treat, although it presents a 5-year mortality of 25%. Though the guidelines (GLs) recommend treatment in specialized centers by physicians specializing in gynecologic oncology, most women are managed by general gynecologists, resulting in differences and discrepancies in clinical management. In this paper we reviewed the literature with the aim of highlighting where the treatment of EC patients requires gynecologic oncologists, as suggested by the GLs. Moreover, we sought to identify the causes of the lack of GL adherence, suggesting useful changes to ensure adequate treatment for all EC patients.

## 1. Introduction

Endometrial cancer (EC) is the most common malignancy of the female genital tract in Western and emerging countries. In 2012, new cancer cases numbered 319,605, and 76,160 cancer deaths were diagnosed worldwide [1,2]. EC pathogenesis can be influenced by a number of factors, including lifestyle, diet, increased obesity rates, hormonal milieu, and age of first pregnancy [1]. EC is usually diagnosed in postmenopausal patients; the highest incidence is at age 70 [3].

In 70% of patients, EC is diagnosed at an early stage because the patients are early symptomatic, and therefore the prognosis is generally favorable, with a 5-year overall survival (OS) of 77%. In contrast, EC patients with advanced or recurrent disease present poor outcomes due to a low response rate to chemotherapy [4,5].

According to the clinical-pathological features, Bokhman introduced a dualistic model to classify EC [6]. Type 1 ECs are endometrioid tumors and represent the most common EC. They are typically diagnosed at an early stage and have a good prognosis. Obesity, diabetes, hyperestrogenism due to anovulatory uterine bleeding, infertility, and late-onset of menopause are well-known causes of type 1 ECs.

Type 2 ECs are non-endometrioid cancers and are rarer and more aggressive. Serous, clear cell, mixed, or undifferentiated carcinomas and carcinosarcomas are included in Type 2 ECs. Type 2 ECs are not associated with hyperestrogenism and may occur in the atrophic endometrium. Type 1 EC patients present a 5-year OS ranging from 75% to 86%, while Type 2 EC patients present a worse 5-year OS, ranging from 50% to 60%. Patient age, International Federation of Gynecology and Obstetrics (FIGO) stage, depth of myometrial invasion, tumor histotype, grade (G), and lymphovascular space invasion (LVSI) are well-known prognostic factors [7,8,9]. New prognostic factors have been investigated [10,11,12,13] to identify tumors with poor outcomes. Prognostic factors drive the staging and adjuvant treatment of EC [12,13,14].

In 2013, the Cancer Genome Atlas molecular classification was proposed to stratify EC patients. The most common EC histotypes have been categorized into four genomic classes: ultra-mutated DNA polymerase ε (POLE) exonuclease tumors (POLE) with a favorable prognosis; microsatellite-instable tumors (microsatellite hypermutated) and low copy-number tumors (microsatellite-stable), both with an intermediate prognosis; and high copy-number tumors (serous-like) with a poor outcomes. Moreover, a subset of ECs diagnosed as high-grade endometrioid cancers harbor copy-number and mutational profiles more similar to those of serous ECs, and in general no mutations (excluding POLE) were identified as unique to any of the four genomic classes [12].

The main EC treatment is total hysterectomy with bilateral salpingo-oophorectomy [14]; thus, EC has been regarded as easy to treat, although 25% of patients die of recurrence within 5 years of the diagnosis [14]. Historically, women are usually managed by general gynecologists (GGs) in low volume centers, resulting in differences and discrepancies in clinical management [15].

In this paper we reviewed the literature on EC treatment to understand at what stage in the management of EC treatment centralization can improve the outcome of EC patients and whether EC patients should be treated by gynecologic oncologists (GOs) in high-volume hospitals [16].

## 2. Pathology

The diagnosis of endometrial cancer is histological and is essential for guiding subsequent treatments. It is obtained on endometrial biopsy, which can be performed in multiple ways: dilation and curettage (D&C), guided hysteroscopic biopsy, vacuum aspiration biopsy random assay, etc.

Expert gynecological pathologists (EP) are essential for choosing the correct treatment and reducing errors in complex diagnoses, especially in the case of frozen sections (FS). Therefore, the pre-operative evaluation of histological slides retrieved from external hospitals by an EP is fundamental. In fact, it has been proven that a second opinion by pathologists sub-specializing in specific fields, and usually working in tertiary care hospitals, can improve the quality of the diagnosis [17,18]. Moreover, expert cytopathologists could also be fundamental in the evaluation of the slides of peritoneal washings or pleural cytology as the diagnostic difficulties can sometimes be challenging [19].

Regarding gynecological cancers, Grevenkamp et al. [20] carried out a specialized histopathology review, analyzing all the ECs diagnosed from 2003 to 2013. Interestingly, EPs corrected 55/565 (9.7%) diagnoses, including 38/565 (7%) cases in which the ESMO risk class became different after the EP review. These major discrepancies (*n* = 38) mainly concerned the tumor histotype (24/55, 44%), grade (10/55, 19%), and myoinvasion (4/55, 7%), while the diagnosis of an EC was not confirmed in 17 cases (10 atypical endometrial hyperplasia [AEH]; 4 endometrial carcinosarcoma; 1 neuroendocrine carcinoma; 1 leiomyosarcoma; and 1 atypical polypoid adenomyoma). Minor discrepancies (not changing the risk class) were found in 214/565 (37.9%) cases, typically involving a different interpretation of low-grade carcinomas (G1 vs. G2; *n* =184) [20].

Diagnostic misinterpretations not only alter the quality of cohorts included in translational research studies but also potentially affect the patient’s care and treatment protocols. Downstaging the tumor, or missing a high-grade/risk tumor component (especially regarding the serous, clear cell, undifferentiated/dedifferentiated, carcinosarcoma, and neuroendocrine carcinoma histotypes) may preclude the patient from additional appropriate treatment, either surgical (lymphadenectomy, peritoneal biopsies, omentectomy, etc.) or chemo-/radiotherapeutic (different drugs, doses, schedules, trials, etc.); conversely, overstaging carcinomas, incorrectly diagnosing high-grade/risk histotypes, or missing a metastatic tumor may lead to avoidable treatments. This concept also holds true in gynecological cancers other than EC. For example, Kommoss et al. [21] conducted a prospective study trying to demonstrate the potential diagnostic discrepancies in ovarian cancer patients enrolled in a third-phase trial: 31/454 (6.8%) cases were misdiagnosed, potentially leading to a different therapeutic management. The two main misdiagnoses consisted of diagnosing: (1) an ovarian cancer instead of a borderline tumor, resulting in a loss of quality of life and fertility caused by radical surgery + chemotherapy; and (2) a primary ovarian cancer instead of a metastasis, leading to unnecessary surgery or inadequate chemotherapy regimens (and preventing more appropriate therapies).

Pre-surgical biopsies and FS examination may also be difficult to interpret in routine practice [22]. For example, another relevant pre-operative question is to determine whether an adenocarcinoma is of endometrial or endocervical origin. Expertise may allow EPs to consider the more unusual histomorphological facets of cancers and to perform the most appropriate immunopanels. Furthermore, the consequences of an incorrect diagnosis weigh on the patient, but avoidable therapies and diagnostic tests also have an impact on the costs of the health system [21]. EPs may choose the most appropriate immunopanels, avoiding unnecessary immunomarkers, and a correct diagnosis can optimize the costs of patient treatment.

## 3. Transvaginal Ultrasound

Useful information about EC invasion can be preoperatively provided by MRI and transvaginal ultrasound examination (TV-US). TV-US is a low-cost, faster, non-invasive diagnostic technique that does not cause significant additional stress for the patient; it aims to identify significant myometrial infiltration (>50%) among low-risk patients.

Recently, a study from the Swedish Gynecologic Cancer Group (SweGCG) reported that the accuracy for predicting deep myometrial invasion was moderate for both TV-US and MRI, and the sensitivity for TV-US was lower than for MRI. SweGCG therefore suggested examining which method for assessing deep myoinvasion is most applicable in each center with regard to examiner expertise and experience [23].

A prospective study of 156 consecutive EC patients reported that MRI had significantly greater specificity than TV-US when assessing deep myoinvasion, but intraoperative FS showed significantly better diagnostic accuracy than both techniques in preoperative imaging. However, the lower performance of TV-US could be explained by the lack of sonographer training in gynecological cancer imaging. [24].

Several recent studies confirmed that the diagnostic performance of TV-US is similar to that of MRI for predicting deep myoinvasion [25,26] and cervical stromal invasion [27] in EC patients. Currently, in referral centers, TV-US has replaced magnetic resonance imaging in the evaluation of myometrial infiltration [28].

When an expert sonographer (ES) performs the preoperative TV-US evaluation of myometrial and cervical stromal invasion, the ultrasound evaluation shows a greater degree of concordance with histopathological characteristics and greater interobserver reproducibility than when the evaluation is done by a gynecologist [28]. Several studies have reported that TV-US showed effectiveness in staging ECs comparable to the results of MRI when the TV-US was performed by an ES [29,30,31,32].

In the last decade, several authors have suggested different approaches for evaluating myometrial and cervical invasion in EC patients. In 1990, Gordon et al. proposed assessing the ratio of the distance between the endometrium–myometrium interface and maximum tumor depth to the total myometrial thickness [33]. In 1992, Karlsson et al. suggested using the tumor/uterine anteroposterior (AP) diameter ratio; a ratio of >0.5 indicated a high risk of deep myoinvasion, with sensitivity of 79% and specificity of 100% [34]. Three-dimensional virtual navigation to measure the tumor-free margin was proposed by Alcazar et al., with a value < 9 mm indicating a high risk of deep myoinvasion, with sensitivity of 100% and specificity of 61% [35].

In 2013, the prospective multicenter study of Mascilini et al. compared different methods of ultrasound myometrial evaluation; they reported that all objective techniques were equally or less reliable with respect to the subjective assessment of cervical and myometrial infiltration.

The best objective measurement techniques to predict deep myoinvasion were the tumor/uterine AP diameter ratio and the minimal tumor-free margin/uterine AP diameter ratio. The distance from outer cervical to lower margin of tumor showed to be the best objective parameter to predict cervical invasion [36]. A systematic meta-analysis by Alcazar et al. [37] compared the diagnostic accuracy of three different methods of TV-US evaluation in detecting deep myometrial infiltration (subjective evaluation, and Gordon’s and Karlsson’s approaches); the authors did not observe any differences in terms of diagnostic performance. They observed significant heterogeneity across studies as regards sensitivity and specificity; a possible explanation could be the potentially low reproducibility of these methods. All these studies involved oncological centers, and all TV-US were performed by the principal investigator of each center. To the best of our knowledge, only one study investigated interobserver reproducibility between ESs and gynecologists.

Compared to gynecologists, the ultrasound examinations performed by ESs were found to have greater sensitivity, specificity, and agreement with the final pathology in the evaluation of cervical stromal invasion but not of deep myometrial invasion. The evaluation of cervical stromal invasion can be very complex as it includes several factors that can often complicate the ultrasound evaluation. It is essential to identify the junction between the lower uterine segment and the upper endocervix, the distinction between cervical and stromal glandular involvement, as well as the distinction between cervical glandular involvement and reactive/non-neoplastic lesions of the endocervical glands.

According to international guidelines, in early low-risk EC patients, expert TV-US and/or MRI and/or FS should be used to assess myoinvasion to decide for lymph node dissection [14].

In conclusion, ultrasound examination can achieve good sensitivity and specificity in assessing the stage of EC as long as it is performed by an ES, as largely demonstrated by several studies. This highlights the importance of the centralization of treatment to high-volume oncological centers to improve diagnosis and treatment in women with EC [38].

## 4. Surgery

The standard treatment of EC patients is total hysterectomy and bilateral salpingo-oophorectomy, and in patients with risk factors, pelvic and para-aortic lymphadenectomy are also required. Since the early 2000s, several studies have sought to evaluate the impact of GOs on EC treatment [39,40,41,42]. GOs managed less than one-half of all EC patients [39] and only 30% of women received lymph node assessment at the time of surgery [39]. Consequently, as many as 70% of patients had postoperative treatment decisions based on missing staging information [39]. On the contrary, staging information was obtained by GOs in a higher percentage of EC patients without increasing morbidity [40,41]. Comprehensive surgical staging made it possible to reduce the use of adjuvant therapy in both early Type 1 and 2 EC [40,41,43,44,45]. Moreover, EC patients treated by GOs are more likely to have disease features that correlate with worse outcomes, including higher grade and stage [41,46]. Despite the higher prevalence of unfavorable disease features, EC patients treated by GOs showed outcomes equivalent to those with more favorable disease characteristics [41]. Unfortunately, although GOs have been increasingly involved with higher risk patients, less than 50% of EC patients are typically referred to a GO [46,47]. Several clinical audits have reported that there was limited tertiary referral to GOs, often with inadequate standards and basic staging procedures performed fully on only one-third of patients [47].

Incomplete staging procedures and inadequate use of adjuvant radiotherapy could be improved to avoid mortality in patients with EC [48]. Another study confirmed that inadequate staging had a significant impact on survival, probably because only 12% of women with EC in Scotland were operated on by a GO. However, this and more recent studies concluded that, despite the fact that centralization of EC patients resulted in accurate staging information, it was not yet clear what impact centralization might have on outcomes [44,49]. Thus, there is no definitive evidence that centralization improves outcomes in EC patients [47]. Low-risk EC may be managed effectively outside referral centers [7]. Although EC patients treated by high-volume surgeons more often present with more aggressive tumors and higher rates of comorbidities, the perioperative surgical and medical complications, as well as the intensive care unit requirements, were all lower in patients who were treated by high-volume surgeons [50,51,52]. Several studies have shown that GOs performed more lymphadenectomies than GGs, and their adequacy rate for lymphadenectomy (number of resected lymph nodes) was higher (compared to GGs) [39,40,41,44,45,47,49]. Moreover, the surgical treatments carried out on early EC patients by GOs and GGs showed similar perioperative outcomes, but operation time and costs were lower when surgery was performed by GOs [42].

However, in patients with low- and intermediate-risk ECs, the use of systematic lymphadenectomy is either contraindicated or controversial, respectively [14]. In the last 10 years, the use of sentinel lymph node biopsy (SLN) has become more widespread and has been included in the international guidelines [14,16]. Systematic lymphadenectomy is associated with increased risk of vessel and nerve injuries, lower extremity lymphedema, thromboembolism, and longer surgery time and hospitalization without improving survival [53,54]. SLN biopsy was proposed as an alternative staging approach, as it may reduce the potential complications associated with lymphadenectomy while still maintaining accurate staging. Moreover, SLN biopsy can reduce the operating room occupancy times for each patient by at least 30–40 minutes, particularly in the case of obese women [16,55,56]. In addition, the more extensive use of the SLN technique can reduce the use of FS, saving additional operation time as well as the pathologists’ time and costs for evaluation. Traditionally, FS were used to drive staging procedures [22,57], but new European Society of Gynaecological Oncology (ESGO) guidelines consider FS to be obsolete because of poor reproducibility and poor agreement with definitive paraffin sections; moreover, FS could interfere with adequate pathological processing [16].

A recent Cochrane meta-analysis found a mean SLN detection rate of 86.9% and the bilateral detection mean rate was 65.4%. Specifically, the SLN detection rate was 77.8% for blue dye alone, and 100% for indocyanine green (ICG) and technetium-99m (^99m^Tc). The rates of positive lymph nodes ranged from 5.2% to 34.4%, with a mean of 20.1%. The pooled sensitivity of SLN biopsy was 91.8% (blue dye alone 95.2%; ^99m^Tc alone 90.5%; ^99m^Tc and blue dye 91.9%; ICG alone 92.5%; ICG and blue dye 90.5%; ICG and ^99m^Tc 100%). This meta-analysis found that the sensitivity of the SLN biopsy was not influenced by the different tracers used or by the different injection sites (subserosal vs. subserosal and cervical only), or by the stage of the EC patients or the different lymph node regions investigated. However, a high-quality intervention study is required to adequately integrate SLN biopsy results with other uterine histopathological and molecular findings useful in the choice of adjuvant therapy [58].

Recent studies have investigated the hypothesis that SLN biopsy could be a valuable technique for diagnosing lymph node metastases in high-risk EC patients. Encouraging results have been obtained, but randomized clinical trials are still needed to confirm this hypothesis [59].

Furthermore, the SLN biopsy technique requires adequate training in order to achieve high detection rates, so every surgeon should know their own detection rates and false-negative rates.

Early experience with SLN showed that a high detection rate can be achieved by performing at least 30 cases of SLN per year, though surgical experience can accelerate the learning curve [60]. A study published in 2020 confirmed that at least 28 procedures are recommended in order for each surgeon to obtain adequate lymph node mapping with the SLN technique [61]. In the same year, another study reported that the plateau of the learning curve for successful bilateral mapping seems to be reached at around 40 cases [62]. The ability to dissect and remove lymph node tissue depends on the experience of the individual surgeon. For every 10 additional procedures performed, there is an 11% increase in lymph node removal and a 5% increase in successful sentinel node identification [62]. However, the SLN procedure is not yet routinely performed in many countries.

Results from a recent survey sent to members of European Society of Gynaecological Oncology, the International Gynecologic Cancer Society, and the Society of Gynecologic Oncology showed that around 50% of 489 physicians from 69 different countries (the majority from Europe and the US) had adopted the SLN procedure. Among the 243 not adopting SLN, 50.2% cited lack of evidence and 45.3% stated that inadequate instrumentation influenced their decision [63].

The SLN should not be considered an easy way to treat EC outside of high-volume centers or by GGs, as the surgeons should be able to perform a systematic pelvic and para-aortic lymphadenectomy in the case of failed detection of SLN localization.

During the last decades, the laparoscopic approach to EC patients has progressively increased, currently becoming the gold standard [14,64]. It is well known that minimally invasive surgery (MIS) is associated with less blood loss and wound infection, shorter hospital stays, as well as a quicker return to normal activity and better cosmesis [4,64,65,66].

Moreover, several studies, including the multicenter, randomized LaCeS trial, showed that the overall costs were lower for laparoscopic than laparotomic hysterectomy [67]; the costs increased with older patient age and comorbidities and were not influenced by the route of surgery alone [68]. Although the surgeons and hospital volumes appear to have little effect on perioperative morbidity, mortality, and resource utilization [66] in EC patients, several studies have shown that laparoscopy was mainly adopted in high-volume centers [44,45,50,69,70]. In particular, laparotomy is still preferred to laparoscopy for high-risk ECs, even if comprehensive surgical staging could be achieved with MIS, resulting in the same outcome as laparotomy [43,44,68]. Similarly, the early use of robotic surgery for the treatment of EC patients occurred in high-volume centers [52,71].

Compared to laparotomy, as well as laparoscopy, robotic surgery is associated with fewer intraoperative complications, lower risk of bleeding and transfusions, lower infection risk, lower postoperative pain, quicker return to daily activities, and lower risk of re-hospitalization. These characteristics are associated with a reduction in hospital costs [72,73,74,75,76]. The robotic approach could increase the percentage of patients undergoing MIS, especially in difficult cases such as morbidly obese patients, where traditional laparoscopy can be particularly demanding [77,78,79]. The robotic approach is characterized by improved visualization and the ability to bend and rotate tools in the same manner of a human hand [71,72].

A recent retrospective study on the economic evaluation of different surgical approaches to EC reported that laparoscopy was associated with the lowest mean cost and open surgery with the highest mean cost per procedure. The robotic approach was associated with rising costs, probably in the case of patients with comorbidities (especially high body mass index). However, the development and adoption of prognostic and risk-stratifying biomarkers in the future may help to choose the best surgical approach for more personalized management of EC patients, as well as for women with other gynecological tumors [68,80].

## 5. Fertility-Sparing Surgery

The 2012–2016 Surveillance, Epidemiology, and End Results (SEER) statistics reported that 6.5% of ECs were diagnosed in women <45 years old [81]. However, the incidence of young women who are diagnosed with EC and its precursor AEH is on the rise, particularly as the rate of obesity increases. In addition, the age of first pregnancy has been progressively increasing in Western countries, so, for these reasons, the issue of fertility-sparing will involve more and more women [82]. Moreover, EC and infertility have common risk factors such as polycystic ovary syndrome, anovulation conditions, obesity, diabetes, and so on. Obesity (body mass index > 25 kg/m^2^) and insulin resistance have been associated with increased EC recurrence rates due to increased resistance to progesterone therapy [83,84,85,86]. Women with AEH or FIGO stage IA G1–G2 EC can be selected for fertility-sparing treatment. Magnetic resonance imaging (MRI) should be performed to evaluate the myometrial infiltration and distant spread. Dilation and curettage (alone or associated with hysteroscopy) should be used to perform endometrial sampling to assess tumor grade and myometrial infiltration. Fertility-sparing treatment includes hormone therapy. Usually, progestogens can be administered, orally or locally, through a progestogen-releasing intrauterine device alone or after hysteroscopic resection of the tumor. However, some physicians have preferred to avoid hysteroscopic resection of the tumor (e.g., polyps or single, small endometrial areas) because it is associated with a greater risk of endometrial synechiae and/or reduction in the implantation rate, with consequent damaging effects on reproductive outcomes [87,88]. A careful selection of patients is mandatory and should be made by a multidisciplinary team in referral centers (recommendation A) [89]. Expert gynecological pathologists should review endometrial samples to confirm the diagnosis (recommendation A) and to differentiate AEH from G1 EC [89]. According to the recommendations of the Task Force of the European Society of Gynecological Oncology, all specimens should be examined by at least two experienced pathologists to improve the accuracy of the final histological diagnosis [89].

An accurate evaluation of the endometrial biopsy is essential in deciding whether or not to perform fertility preservation. Furthermore, the differentiation of AEH from G1 EC is crucial since the response rate of these two entities to hormone therapy is different. Patients with AEH showed a 66% response rate to therapy, while patients with EC G1 showed a 48% response rate with more frequent disease persistence [90].

Unfortunately, the literature reports an interobserver and intraobserver discrepancy rate of approximately 40% in the distinction between AEH and EC [91]. When the EC shows an excellent response rate, misdiagnosed AEH should be suspected. The expression of the progesterone receptor (PR) could be useful in predicting the response to hormonal therapy, but it is not mandatory; in fact, progestin therapy can also be administered in the case of tumors that do not express PR [89]. On the contrary, PR-subtype expression should be evaluated in the case of tumor resistance to treatment. Resistance to treatment might be due to reduced expression of the PR subtype B in EC cells induced by medroxyprogesterone acetate treatment [92]. However, contraindications to medical therapy or pregnancy should always be excluded. A fertility specialist should evaluate each patient’s reproductive capabilities before starting treatment and plan a strategy to achieve a pregnancy as soon as possible [93,94]. Nulliparity and sterility are common in EC patients younger than age 45, with about 61% of EC patients affected [95]. However, all EC patients undergoing fertility-sparing treatment should be advised to undergo assisted reproductive technology (ART) to achieve pregnancy in the shortest time possible [88]. Pregnancy itself represents high-dose hormone therapy since the progestin concentration in the blood is much higher than that normally given during therapy [87]. In addition, it may be possible to plan standard treatment at the end of the desire for parenthood [87]. Moreover, a careful and accurate follow-up should be performed in order to change strategy in a timely manner in the case of unsuccessful therapy.

## 6. Multidisciplinary Evaluation

High-volume centers are usually characterized by work in a multidisciplinary team involving other specialists, such as general surgeons, anesthesiologists, medical oncologists, psychologists, gynecological pathologists, radiation oncologists, palliative care specialists, radiologists, geneticists, and researchers. Multidisciplinary evaluation of EC patients guarantees the most appropriate management of these patients to provide them with the best care, according to practitioner experience and the latest guidelines. According to the ESMO guidelines [14], in intermediate-risk EC patients (stage I endometrioid, grade 1–2, ≥50% myoinvasion, LVSI negative) adjuvant brachytherapy (BRT) should be administered (level of evidence: I; strength of recommendation: B), but for patients aged <60 years, no adjuvant treatment is an option (level of evidence: II; strength of recommendation: C). In high–intermediate risk EC patients (stage I endometrioid, grade 3, <50% myoinvasion, regardless of LVSI status; or stage I endometrioid, grade 1–2, LVSI unequivocally positive, regardless of depth of invasion) different adjuvant therapies are given based on lymph-node status. In the case of negative nodes after surgical nodal staging, adjuvant BRT is recommended to decrease vaginal recurrence (level of evidence: III; strength of recommendation: B), but no adjuvant therapy is an option (level of evidence: III; strength of recommendation: C). In the case of absent surgical lymph-node staging, adjuvant external beam radiation therapy (EBRT) is recommended for LVSI which is unequivocally positive to decrease pelvic recurrence (level of evidence: III, strength of recommendation: B). Adjuvant BRT alone is recommended for grade 3 and LVSI negative to decrease vaginal recurrence (level of evidence: III; strength of recommendation: B). Systemic therapy is of uncertain benefit (level of evidence: III; strength of recommendation: C). In high-risk EC patients (stage I endometrioid, grade 3, ≥50% myoinvasion, regardless of LVSI status). In the case of negative nodes after surgical nodal staging, adjuvant EBRT with limited fields should be considered to decrease locoregional recurrence (level of evidence: I; strength of recommendation: B). Adjuvant BRT may be considered an alternative to decrease vaginal recurrence (level of evidence: III; strength of recommendation: B). Adjuvant systemic therapy is still debated (level of evidence: II; strength of recommendation: C). In the case of absent surgical lymph node staging, adjuvant EBRT is generally recommended for pelvic control and relapse-free survival (level of evidence: III; strength of recommendation: B) and sequential adjuvant chemotherapy may be considered to improve progression-free survival (PFS) and cancer-specific survival (CSS) (level of evidence: II; strength of recommendation: C). Chemotherapy and EBRT should be administered in combination rather than either treatment modality alone (level of evidence: II; strength of recommendation: B).

In stage II, grade 1–2, LVSI-negative EC patients with negative node after surgical nodal staging, BRT is recommended to improve local control (level of evidence: III; strength of recommendation: B). In the case of stage II, grade 3 or LVSI unequivocally positive EC patients, limited field EBRT is recommended (level of evidence: III; strength of recommendation: B) and a BRT boost may be added (level of evidence: IV, strength of recommendation: C). In these EC patients, chemotherapy use is still debated (level of evidence: III; strength of recommendation: C). In the case of stage II EC patients without surgical nodal staging, EBRT is recommended (level of evidence: III; strength of recommendation: B) and a BRT boost may be added (level of evidence: IV; strength of recommendation: C). In the case of stage II, grade 3 or LVSI unequivocally positive EC patients sequential adjuvant chemotherapy should be administered (level of evidence: III; strength of recommendation: B). In patients with high-risk, stage III EC and no residual disease EBRT is recommended to decrease pelvic recurrence and PFS (level of evidence: I; strength of recommendation: B) and to improve OS (level of evidence: IV; strength of recommendation: B). Chemotherapy should be administered in combination with EBRT to improve PFS and CSS (level of evidence: II; strength of recommendation: B). In high-risk, Type 2, full staged EC patients, chemotherapy should be administered (level of evidence: III; strength of recommendation: B). In the case of serous and clear, full-staged EC patients at stage IA with LVSI negative, BRT alone could be administered without chemotherapy (level of evidence: IV; strength of recommendation: C), while in stage ≥IB, EBRT in addition to chemotherapy should be administered, especially for node metastasis (level of evidence: III; strength of recommendation: C). In carcinosarcoma and undifferentiated EC patients, chemotherapy is always recommended (level of evidence: II; strength of recommendation: B), and EBRT could be administered (level of evidence: III; strength of recommendation: C). In the case of EC patients with advanced (defined as bulky FIGO stage IIIA-IV) or relapsed disease, surgical therapy should only be considered if complete gross resection of the tumor can be achieved (level of evidence: IV; strength of recommendation: C). Patients with advanced or recurrent EC are generally candidates for systemic palliative therapy. Several factors such as histopathological and clinical characteristics guide the choice between chemo or hormone therapy. In the case of isolated vaginal recurrence of early-stage EC, radiotherapy can be given to improve local disease control (level of evidence: III; strength of recommendation: A). Surgery or chemotherapy could be used in the case of vaginal or bulky pelvic-node recurrence before administering radiotherapy (level of evidence: V; strength of recommendation: C). Additionally, re-irradiation could be considered in selected patients (level of evidence: V; strength of recommendation: C).

Recently, molecular tumor boards have also been introduced into routine practice. As reported in the new ESGO guidelines [16], molecular biology plays an increasingly pivotal role in supporting different therapeutic decisions and suggesting prevention strategy.

The new ESGO guidelines [16] suggest considering anti-PD1-based immunotherapy with pembrolizumab as second-line therapy in recurrent patients with MSI/MMRd EC.

The combination of pembrolizumab and the multi-tyrosine kinase inhibitor lenvatinib could be considered for second-line treatment of microsatellite-stable carcinomas (level of evidence: III; strength of recommendation: B) [16].

Moreover, triage for germline mutation analysis is strongly required to identify patients with Lynch syndrome. Immunohistochemistry to evaluate deficiency in the maladjustment repair–system proteins and/or molecular tests to evaluate microsatellite instability should be performed in all ECs, regardless of histology [III; B].

In high-grade tumors, the molecular classification [IV; B] must always be performed, while in low- and intermediate-risk ECs with low-grade histology, the analysis of the POLE mutation can be omitted [IV; C].

In POLE-mutated EC patients at stage I–II, adjuvant treatment should be omitted [III; A] [96,97]. A recent study of 359 EC patients with the POLE mutation included 294 (82%) patients with pathogenic mutations who showed a better outcome than patients with the non-pathogenic POLE mutation.

Interestingly, in EC patients with pathogenic POLE mutation, only the stage influences the risk of progression/recurrence or death, while all other known prognostic factors show no significant effect. Furthermore, in the case of relapse, POLE mutation patients show high salvage rates with a good prognosis. However, the prognosis of these EC patients is not affected by adjuvant therapies [98].

In contrast, adjuvant combination therapy is recommended in EC stage I–III patients with p53 mutant carcinoma because it is associated with a statistically significant survival advantage [99]. However, adjuvant therapy is not recommended in EC patients with p53abn carcinomas restricted to a polyp or without myoinvasion [III; C] [99,100].

## 7. Hub-and-Spoke Model

EC is the most common cancer in Western and emerging countries. The “Westernization” of developing countries is progressively leading to an increase in risk factors for EC. The costs for EC treatment are presumably going to increase, including those due to expensive MIS approaches such as robotic surgery (particularly in obese patients), the use of technology for SLN staging, the molecular tests needed for EC classification, and follow-up visits. EC follow-up absorbs many resources of the health system without achieving a significant improvement in overall survival even when performed intensively in EC with the highest risk of recurrence [101].

Therefore, the identification of a correct and effective therapeutic path becomes essential both for the use of effective treatments and for the correct use of resources, especially in the historical post-COVID-19 pandemic period, as resources are increasingly becoming scarce and valuable in terms of money to invest, the time available, and the establishment of spaces for the treatment of these patients.

In the literature, the centralization of EC patients seems to be associated with accurate preoperative assessment, frequent use of MIS, comprehensive surgical staging, proper use of adjuvant therapy, and a multidisciplinary approach (Table 1) [102].

Centralization also reduces the learning curve of complex surgical procedures and the individual case load of each surgeon, thus facilitating innovation and overcoming the learning curve problems encountered in low-volume centers.

Unfortunately, it is not always possible to perform surgery in high-volume hospitals, thus the hub-and-spoke model has been advocated. Several clinical audits showed that this model can guarantee uniform and adequate treatment to EC patients. It is crucial to select patients to centralize and patients to treat in spoke hospitals. Selection should be accurate and timely. Long surgical waiting times can worsen prognosis [102] due to disease progression or difficulty in accessing treatment. The hub-and-spoke model should ensure fast access to care for all EC patients; surgical waiting time ≥ 8 weeks is known to worsen the 5-year survival of low-risk EC patients [103,104]. In fact, surgical waiting times in referral centers are generally longer than in community hospitals [105]. This might be due to the characteristics of the patients referred to hub centers who have to be studied accurately because of more aggressive disease or severe comorbidities.

A possible EC pathway should include centralized pathological evaluation of all endometrial biopsies by experienced gynecologic pathologists, and a revision of the external EC diagnosis should be requested before proceeding to the preoperative workup. Type 2 ECs are treated at the hub center, and Type 1 ECs undergo ultrasound evaluation at the hub center to assess myo- and cervical invasion. Low-risk EC patients should be treated at spoke centers while all other EC patients should be centralized. All advanced EC patients, patients selected for fertility sparing, and all frail patients or patients with major comorbidities should be treated at the hub center. In low-risk ECs, FS could be avoided, systematic lymphadenectomy should be avoided, and SLN procedures can be used in the case of an experienced surgeon, although it is not a mandatory procedure. In high-risk ECs, adequate surgical staging should be performed and FS could be avoided. SLN procedures can be used according to the Mayo Clinic SLN algorithm for high-risk EC [106].

All final pathological diagnoses are performed at the hub center and all EC patients are managed by gynecological tumor board members. EC patients who are to receive radiotherapy should be centralized, while adjuvant chemotherapy therapy could be provided by spoke centers. According to clinico-pathological and molecular features, all high-risk EC patients and all EC patients who have received adjuvant therapy should be followed up at the hub center by GOs. All low-risk EC patients who did not receive adjuvant therapy could be followed up at spoke hospitals or by telephone follow-up [107] (Figure 1).

## 8. Conclusions

Although most ECs are diagnosed at an early stage and a total hysterectomy with bilateral adnexectomy by a GG may be adequate treatment, some steps of EC diagnosis and treatment cannot be separated from evaluation by a gynecological oncology expert. These crucial assessments can be ensured by high-volume centers (hubs), where specialists can acquire and maintain the appropriate skills.

In hub centers, the selection of patients to be sent to spoke centers, the selection of patients to be sent for adjuvant therapy, the selection and treatment of patients to be subjected to fertility sparing, and the selection of the type of follow-up should be made.

The hub should also ensure adherence to clinical trials for patients managed in spoke hospitals and should promote data collection and clinical research.

Finally, the hub center should also promote the training of spoke-and-hub center specialists dedicated to EC treatment and should share guidelines and decision-making processes to ensure effective, timely, and consistent EC care.

## Figures and Tables

**Figure 1 biology-11-00768-f001:**
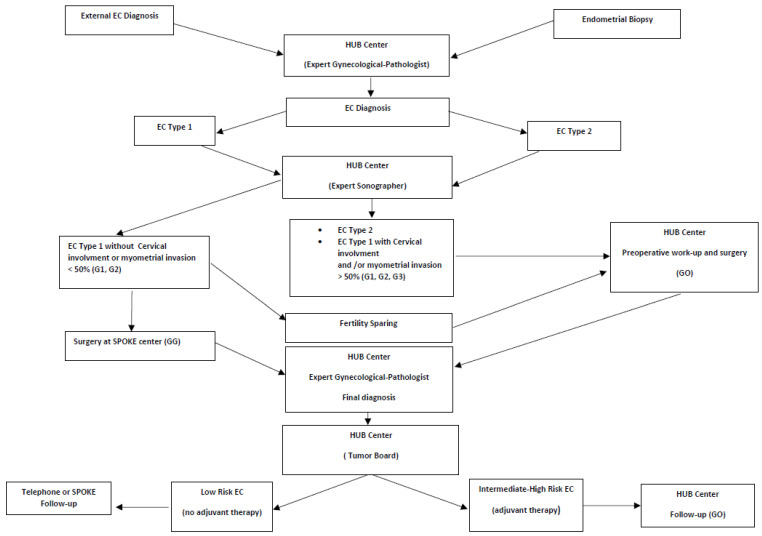
Preoperative work-up and centralization of endometrial cancer patients.

**Table 1 biology-11-00768-t001:** Advantages and disadvantages of centralizing endometrial cancer patients.

Author, Year	Country	Advantages	Disadvantages
Crawford SC et al. [48], 2001	Scotland	Surgical staging, Lymphadenectomy, Adjuvant therapy	-
Roland PY et al. [40], 2003	United States	Complete staging, Experience of gynecologists/oncologists, Minimization of the potential morbidity associated with adjuvant radiation	Geographical difficulties for access to the center
Macdonald OK et al [41], 2005	United States	More appropriate adjuvant therapy	-
Parkin DE et al. [49], 2006	Scotland	Complete staging, Correct selection of patients for adjuvant treatment (lymphadenectomy or radiotherapy), Multidisciplinary team	-
Hoekstra A. et al. [42], 2006	United States	Operative time and cost, Experience of gynecologists/oncologists, Appropriate follow-up	-
Savelli L et al. [32], 2008	Italy	Lower costs related to the presence of TVS performed by expert specialists	-
Mandato VD et al. [45], 2012	Italy	Appropriate pre-surgical assessment, Multidisciplinary evaluation, Appropriate surgery treatment (laparoscopy, lymphadenectomy), Presence of expert pathologists	-
Wright JD et al. [50], 2011	United States	Improved perioperative surgical/medical complications and ICU	-
Greggi et al. [69], 2014	Italy	Optimal surgical treatment for high-risk cases	No benefit for low-risk cases
Chan JK et al. [71], 2015	United States	Robotic surgery, Experience of gynecologists/oncologists, Cost-effectiveness.	Socio-economic barriers could delay the diagnosis and results
Eriksson et al. [29], 2015	Europe	Improved preoperative ultrasound staging (ultrasound experts)	-
Doll KM et al. [46], 2016	United States	Appropriate surgery treatment (lymphadenectomy), High survival for patients who, after centralization, undergo chemotherapy in small centers	Geographical difficulties for access to the center for racial/ethnic minorities who are more likely to live in close proximity to gynecologic oncologists
Seagle BLL et al. [100], 2017	United States	Standardization of adjuvant therapy	-
Green RW et al. [38], 2018	Europe	Superior diagnostic modalities (ultrasound experts)	-
Spoor E. and Cross P. [18], 2019	UK	Greater diagnostic accuracy (expert pathologists)	-
Knisely A et al. [105], 2020	United States	-	Increased travel distance may adversely affect care (limits or delayed access to care)
Mandato VD et al. [22], 2020	Italy	Expert pathologists, Appropriateness of adjuvant therapy	
Mandato VD et al. [44], 2021	Italy	Fewer peri- and post-operative complications, Expert pathologists, Laparoscopy, Appropriateness of adjuvant treatment	-

TVS: transvaginal ultrasound; ICU: intensive care unit.

## Data Availability

Data sharing not applicable.

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
