# Peer review of "Should Endometrial Cancer Treatment Be Centralized?"

_biology, 2022, doi:10.3390/biology11050768_

Round 1

Reviewer 1 Report

The article is titled "Should endometrial cancer treatment be centralized? The points I pointed out in the previous issue have been properly corrected, making it very easy to read and understand.
I thought it should be published in this form.

Author Response

We want to thank the reviewer for his kind opinion

Reviewer 2 Report

Dear Authors,

I found the manuscript very interesting and clear in its message. I think it is very important to centralize EC treatment in referral centres mainly for patients requiring fertility sparing management or for decision about adjuvant therapy when it is needed. However before publication it needs a deep English language review also in teminology (i.e.  page 3, line 150 "deep myocardial invasion"). 

Author Response

We want to thank the reviewer for his kind suggestions. we completely revised the English of our manuscript. we hope our article is now suitable for publication.

Reviewer 3 Report

The authors reviewed the current studies to understand whether the treatment of EC cancer should be served for the gynecologic oncologist and if EC patients should be centralized in high‐volume hospitals. The authors suggested a possible network to treat EC patients to ensure an effective staging and treatment. I think the author's opinions are helpful for the management of EC patients. 

In terms of language and the organization of the context, there are serious problems in the manuscript. The authors should read thoroughly a couple of times, pay attention and spend effort on writing. Otherwise, no one will read the paper if there are full of typos and mistakes in language.  For examples,

  1. 'gynecologist oncologist'  should be gynecologic oncologists
  2. 'we try to suggest e possible network',   'e' should be 'a'
  3.  'guidelines (GLs)' ,   It doesn't make sense to use abbreviation for one word.
  4.  'ultra‐mutated DNA polymerase ɛ (POLE) exonuclease tumors (POLE) with a favorable prognosis,'   I am not sure what happens to this sentence.   
  5. 'than both. techniques. preoperative imaging. However, the lower performance of TV‐US could be explained by the lack of training in gynecological cancer imaging of sonographers. [24].'    Lots of periods in one sentences.

  6. 'Actually, in referral centers, US‐TV has replaced magnetic resonance imaging in the evaluation of myometrial infiltration [28]' Why can this sentence stand alone as a separate paragraph?   also many other cases like this in the manuscripts
  7. 'All low risk EC patients that did not receive adjuvant therapy could be follow up at spoke hospitals or by telephone follow‐up [108]. (Figure 1).'  Extra period is in the sentence.   Where is Figure 1 and its legend?
  8. 'These crucial assessments can be ensured by high volume centers (HUBs),'    There are both capitalized HUB vs small case hub in the manuscripts.  Are they the same? What abbreviation should be for 'high volume centers'?  not HVCs?

Author Response

We want to thank the reviewer for his kind suggestions. we completely revised the English of our manuscript. We hope our article is now suitable for publication.

This manuscript is a resubmission of an earlier submission. The following is a list of the peer review reports and author responses from that submission.

Round 1

Reviewer 1 Report

Thank you for giving me the chance for reviewing this manuscript.

The title should consider that the paper is actually not a review. The authors did not provide any methods or reviewing criteria. There are no data about how much papers were reviewed, what kinds of papers and what were the inclusion and exclusion criteria. The conclusion is not supported by the results and written as a part from discussion even with new citations!!

Reviewer 2 Report

Major revision

P.2 Line 75-Surgery is the main treatment for EC, but it is not an easy treatment. In my opinion, surgical treatment is good for only FIGO stage IA and, no LSVI, and Type 1 cases. Otherwise, multidisciplinary treatment with chemotherapy or radiotherapy is required in addition to surgery. In your institution, do you end up treating patients with surgery alone? If so, it is no wonder that the prognosis is poor. Also, These contents may differ from the descriptions with the following surgical items.

According to many data and experiences, the recurrence and death cases of uterine cancer are how to treat patients with stage IB or higher and type 2. 5-year survival rate of stage IA and type 1 patients is more than 95%, and they can survive surgery without adjuvant therapy. These patients do not need to be treated in a high volume center. Minimally invasive surgery such as robotic surgery can also be performed. In other words, how to manage other patients may contribute to improving the prognosis of uterine cancer patients. I think that this manuscript should focus on this.

There is no mention of chemotherapy or molecular targeted therapy. We believe that the disease cannot be cured by surgery alone and that multidisciplinary treatment is necessary.

Also, Ultrasound has been described. A more accurate assessment of muscle layer invasion can help determine the stage of progression. MRI is more accurate in assessing myometrial invasion.

The conclusion is too long and it is hard to judge what the author is trying to say. The conclusion also contains content that should be DISCUSSION, and in the end, it is not clear from this paper what new ideas or treatments should be used to improve the prognosis.

Minor revision

P2. Line 51-56 The cited manuscript is over 10 years old and outdated.

P2.70-The story has suddenly changed and is very difficult to read. The flow with the relation to the prognosis by Type in the immediately preceding sentence is interrupted. Please consider reorganizing it.

P.2 Line77-Isn't the data very old and also only about Scotland? It is hard to say that this is necessarily the case around the world.

Reviewer 3 Report

This is a narrative literature review on an interesting question with respect to standars of care of endometrial cancer.

Based on the findings of their review the authors advocate a `hub and spoke´ model, in which patients should be assessed and high risk cases should be allocated to high volume centers.

After reading the review, I have the following questions and suggestions:

  • The outline of this review does not make sense to me. To start with 'surgery', then 'fertility surgery', 'Sentinel node biopsy' followed by pathology and transvaginal ultrasound is rather confusing. The authors should follow the patient path from diagnosis to adjuvant or palliative treatment and aftercare. All surgical aspects should be in one paragraph probably structured by subheadings
  • As the authors want to draw conclusions for the current situation it is questionable, if surgical studies from the early 2000s (prior to the widespread use of MIS) will actually contribute to answer the questions. Especially the use of systematic lymphnode assessment has been subject to an ongoing debate, and comprehensive staging including systematic node dissection, per se can no longer be considered as an indicator quality of care. This should be discussed. The true advated is, that GOs perform a more comprehensive (better) lymphadenectomy in those cases, in which medical risk is high and detection of positive lymphnodes is crucial
  • another important factor in favour of centralization is the learning curve of complex surgical procedure and the individual case load of each surgeon providing the chance of easily overcoming learning curve issues found in low volumen centers (eg. Melendez et al. 1997).
  • Table 1 and 2 are not very appealing to the reader. I think it makes sense to summarize common findings and to annotate them with the reference number
  • in the introduction section the authors elaborate on the outdated Bokham classification. They should introduce the modern molecular classification as proposed by Katdoth et al.
  • I think the most important point of 'fertility sparing surgery' is not the surgical but the interdisciplinary part and therefore to my mind should be adressed unter multidisciplinary evaluation. The crucial part is the diagnosis by an expert pathologist and the availability of fertility specialist (not the D and C...)
  • Reference 16 and 70 are identical

Reviewer 4 Report

The work is interesting and the topic is relevant, but does not define a real solution to the question of the study and adequate conclusions. Conclusions should be redefined because currently they seem not supported by the literature analyzed and reviewed. The paper message is quite contradictory. The management and treatment of "non-complex" patient could be done in spoke hospitals to reduce surgical waiting times in hub, but from the analyses of the literature reviewed and discussed by authors emerges that an optimal and accurate preoperative assessment and the adequate treatment with multidisciplinary approach could be done only with centralization of patients in the hub hospital both in early and advanced disease. Based on these assumptions, it does not seem appropriate to propose the management of "less complex cases" on spoke hospital with the only objective to reduce surgical waiting time, because this would entail a risk of under- or overtreatment of the patient, with negative effects on quality of life and/or survival rate. It should be made a proposal for an accurate and efficient management of the patient affected by endometrial cancer; who makes the correct patient selection? An GO or a GG? In a hub or in a spoke hospital? What should be the expertise of GG in the spoke center? Finally, the high number of self-citations appears to be quite inappropriate for contribution they give to the work.